# Apicortin, a Constituent of Apicomplexan Conoid/Apical Complex and Its Tentative Role in Pathogen—Host Interaction

**DOI:** 10.3390/tropicalmed6030118

**Published:** 2021-06-30

**Authors:** Ferenc Orosz

**Affiliations:** Research Centre for Natural Sciences, Institute of Enzymology, Magyar Tudósok Körútja 2, 1117 Budapest, Hungary; orosz.ferenc@ttk.hu

**Keywords:** apicortin, *Plasmodium*, *Toxoplasma*, p25alpha domain, DCX domain, apicomplexa, conoid

## Abstract

In 2009, apicortin was identified in silico as a characteristic protein of apicomplexans that also occurs in the placozoa, *Trichoplax adhaerens*. Since then, it has been found that apicortin also occurs in free-living cousins of apicomplexans (chromerids) and in flagellated fungi. It contains a partial p25-α domain and a doublecortin (DCX) domain, both of which have tubulin/microtubule binding properties. Apicortin has been studied experimentally in two very important apicomplexan pathogens, *Toxoplasma gondii* and *Plasmodium falciparum*. It is localized in the apical complex in both parasites. In *T. gondii*, apicortin plays a key role in shaping the structure of a special tubulin polymer, conoid. In both parasites, its absence or downregulation has been shown to impair pathogen–host interactions. Based on these facts, it has been suggested as a therapeutic target for treatment of malaria and toxoplasmosis.

## 1. Name

Apicortin was identified in silico, in 2009, as a characteristic protein of apicomplexans [1]. It combines a partial p25alpha domain with a DCX (doublecortin) one. Based on its occurrence and one of its characteristic domains, it was termed apicortin.

## 2. Occurrence

Apicortin, when identified, was shown to occur in apicomplexan parasites and in the placozoan animal, *Trichoplax adhaerens* [1]. The apicomplexan genomes known then contained it without exception. This situation practically has not changed since then; this statement is valid for the newly sequenced genomes and transcriptomes of apicomplexans as well. The only exception is the Apicomplexa with the smallest genome, *Babesia microti* [2].

Later it has been found that apicortin also occurs in chromerids, the recently discovered [3,4], free-living cousins of apicomplexans [5]. This is not surprising, given the phylogenetic proximity and the structural similarity of these phyla. Unlike apicomplexan species, both *Chromera velia* and *Vitrella brassicaformis* have three apicortin paralogs. Apicortin has not been found in other related phyla of the Alveolata superphylum, although its remnant is present in the genome in the case of Perkinsozoa [6]. However, the very recently published draft genome of *Perkinsus olseni* contains hypothetical protein(s) possessing both p25alpha and DCX domains (KAF4710163, KAF4750811) [7].

It has also been revealed that some primitive fungi also possess this protein; first it was shown in the cases of *Spizellomyces punctatus* [6] and *Rozella allomycis* [5]. Later, a systematic examination of fungal genomes showed that the flagellated fungi contain apicortin almost without exception; and it is present even in a non-flagellated but also deeper branching clade (Endogonomycetes) [8].

Apicortin is one of the most abundant proteins of *T. adhaerens* [9]. This is the only animal that possesses apicortin [10]. Animal draft genomes and transcriptomes contain sometimes nucleotide sequences, contigs and TSAs (transcriptome shot-gun assemblies), homologous to apicortin, but they have been shown to be contaminations from parasitizing apicomplexans, based on sequence similarities and GC ratios [11,12]. In the case of *T. adhaerens*, this option was excluded by phylogenetic analysis [8]. Several authors suggested that the presence of frequent contaminants could serve as a basis for the identification of hitherto unknown apicomplexans and, in general, parasites [13,14,15]. The origin of a few plant and algal apicortin-like nucleotide sequences needs further investigations. My preliminary data suggest that contamination of apicomplexan origin can be ruled out but not that of fungal ones.

Strong correlation between the presence of the p25alpha domain and that of the eukaryotic flagellum was suggested before the identification of apicortin [16]. With very few exceptions, each flagellated organism contains p25alpha domain-containing proteins; on the other side, in non-flagellated species, the p25alpha domain generally does not occur [8,10]. The protein that contains the p25alpha domain varies depending on the phylum; e.g., it is the so called “long-type” TPPP in animals (except *T. adhaerens*) [10,17], a fungal-type TPPP and apicortin in flagellated fungi [8], while the “short-type” TPPP and apicortin are found in apicomplexan species [10]. With the exceptions of two non-flagellated fungi and some apicomplexans, apicortin can be found only in species which are flagellated, at least in some life stage.

## 3. Domains

Apicortin belongs to a eukaryotic protein superfamily, the TPPP-like proteins, characterized by the presence of the p25alpha domain (Pfam05517, IPR008907), and named after the first identified member, TPPP/p25, which exhibits microtubule stabilizing function [10]. TPPP stands for tubulin polymerization promoting protein [18,19] and was first identified by Takahashi et al. as p25 protein [20]. Full-length p25alpha domain consists of about 160 amino acids; however, that of the apicortin is incomplete, containing only the last 30–40 amino acids (a “partial p25alpha domain”). Importantly, the tubulin/microtubule binding amino acid sequence is located in this part of the domain [21,22,23,24]. This is the most conserved part of the domain, which contains a characteristic “Rossmann-like motif”, GXGXGXXGR [10,17].

However, apicortins possess another characteristic domain, the DCX one (Pfam03607, IPR003533) [1]. The DCX (doublecortin) domain is named after the brain-specific X-linked gene doublecortin [25]. It is a structural domain, which generally appears in duplicate as two tandemly repeated 80 amino acid regions (N- and C-terminal type DCX domains), but proteins containing only one DCX-repeat have also been identified [26,27]. This domain is also known to play a role in the stabilization of microtubules [25,26]. The two domains, the partial p25alpha and the DCX, have some homology: 22% identity and 30% similarity [28].

## 4. Primary, Secondary and Tertiary Structure

Apicortins consist of four structural units (Figure 1): (i) a long, disordered N-terminal domain; (ii) a partial p25alpha domain; (iii) a disordered linker region; (iv) and a DCX domain [5].

The N-terminal part consists of about 70–90 amino acids in apicomplexan apicortins, similarly to the apicortins of the few non-flagellated fungi; this part is 40 amino acid long in one of the *C. velia* apicortins (Cvel_6797) while in other species (placozoa, chromerids, flagellated fungi) it is absent. The N-terminus of the proteins is rather different among the various apicomplexan species. It is not conserved among the species belonging to different classes of Apicomplexa, or the various genera of the same class, or even among the various *Plasmodium* subgenera. The extra N-terminal regions are significantly similar in the two fungal apicortins possessing it. Concerning the secondary structure, the N-terminal region of apicortins was shown to be highly disordered by predictor programs based on different principles [5,7,8].

The partial p25alpha domain, in general, is the most conservative part of the protein (Figure 2), however, there are two exceptions. *R. allomycis* and *Plasmodium* orthologs present in species infecting mammals (but not birds) lack the final part of this domain, which includes the Rossmann-like motif (Figure 2). Otherwise, the sequences are very similar, independently of whether the protein can be found in an apicomplexan species or not. It was shown by Leung et al. [28] that this region of the protein has an outstanding role in tubulin binding.

In the linker (interdomain) region, which has also been predicted to be unstructured, similarity is much lower between protist and non-protist apicortins. Moreover, this part of *Plasmodium* proteins also differs somewhat from those of the other apicomplexan orthologs.

In the DCX domain, the overall similarity is somewhat lower between the two groups than in the case of the partial p25alpha domain; however, there is no exception: the similarity occurs through the whole domain in all orthologs. Phylogenetic analysis showed that DCX domains of apicortins are clustered neither with C- nor N-terminal type domains but form a separate group [5].

## 5. Phylogenetics

Phylogenetic analysis clearly showed that apicomplexan apicortins form a monophy-letic group and are well separated from opisthokont (fungal and placozoan) and chromerid apicortins (Figure 3) [7,8]. One pair of the chromerid apicortins (“Chomerida1” in Figure 3—Vbra_15441 from *V. brassicaformis* and Cvel_6797 from *C. velia*) is a sister group to them, the other 2-2 chromerid homologs are significantly different.

## 6. Apicomplexan Apicortins

Apicortin is a characteristic protein of Apicomplexa. It is present in both classes, Aconoidasida and Conoidasida, and in each order and family whose species are fully sequenced (Appendix A). According to the NCBI website, 66 genomes of apicomplexan species have been fully sequenced, 65 of which contain apicortin. The only exception, *B. microti* has a significantly decreased genome which is the smallest one in the phylum [2]. However, it contains a p25alpha domain containing protein, a short-type TPPP (XP_012649535).

Not surprisingly, the genomes of most *Plasmodium* species have been sequenced (21), due to their epidemiological significance, followed by 15 sequenced *Cryptosporidium* species genomes (Appendix A). Apicomplexans are obligate parasites causing serious illnesses in humans and domestic animals [30]. Species in the genus *Plasmodium* cause malaria, from which over 200 million people suffer each year. The official deaths according to WHO were about 400,000 both in 2018 and 2019 [31]. Other members of the phylum Apicomplexa are responsible for animal sicknesses, such as coccidiosis and babesiosis, resulting in significant economic burden for animal husbandry. *Cryptosporidium* causes cryptosporidiosis in humans and animals, *Theileria* causes tropical theileriosis and East Coast fever in cattle, and *Toxoplasma* causes toxoplasmosis in immunocompromised patients and congenitally infected fetuses. The disease-causing parasites belong to the orders Haemosporida, Piroplasmida and Eucoccidiorida, which explains the bias in genome sequencing (Appendix A).

## 7. Life Stages of Apicomplexans

The life cycle of apicomplexan parasites is characterized by three processes: sporogo-ny, merogony and gametogony [30]. Sporogony consists of an asexual reproduction resulting in the production of sporozoites. Sporozoites are an invasive form that invade cells and develop into forms that undergo another asexual replication, merogony (A proportion of the sporozoites from some *Plasmodium* species go through a dormant period (hypnozoite) instead of the asexual replication [32]). The resulting merozoites are also invasive forms which can invade cells and initiate another round of merogony. Merozoites are known by many different names depending on the species. In *Plasmodium*, the merozoites invade erythrocytes, where the parasite undergoes a trophic period and forms a trophozoite, which ingests the host cell cytoplasm [33]. This period is followed by an asexual replication, the schizont stage. The host erythrocyte releases the merozoites after its rupture. These merozoites invade new erythrocytes and initiate another round of schizogony. Instead of asexual replicative stage, merozoites can develop into gametes through a process called gametogony (gametogenesis). In sexual reproduction, the gametes fuse to form a zygote which first develops into an ookinete (a motile invasive stage) and then becomes an oocyst where sporogony takes place [30].

## 8. Structural Features of Apicomplexa: The Apical Complex and the Apicomplexan Cytoskeleton

Apicomplexans are an ancient and diverse phylum with peculiar cell biological properties. The defining character of the Apicomplexa is the unique assembly of organelles termed the apical complex, from which the name of the phylum was derived (later, it became clear that chromerids and perkinsids also contain an apical complex). The complex is an assembly of membranous and cytoskeletal elements at the apical end of the cell that performs essential functions in both the invasion of the host cells and in the replication of the parasite. The apical complex can be divided into three structurally distinct groups of components [34]: the apical cap that comprises the most apical part of the inner membrane complex; the conoid located within the apical polar ring; and the secretory organelles termed micronemes and rhoptries that deliver the enzymes necessary for the establishment and maintenance of host cell infections. Haemosporidia and Piroplasmida lost the conoid although they retained an apical polar ring.

Many of the distinct traits are related to the unique cytoskeletal elements of these parasites [35]. Apicomplexans possess discrete populations of microtubules. Spindle microtubules are similar to those found in other organisms, made of 13 protofilaments of α- and β-tubulin dimers arranged in a hollow tube [36]. The spindle microtubule population is associated with centrioles, which have an unconventional form consisting of a central single microtubule surrounded by nine singlet microtubules, in contrast to the conventional 9 + 0 structure of nine triplet microtubules [35]. The apical polar ring serves as the microtubule-organizing center for the subpellicular microtubules composed of canonical microtubules. Their number varies among apicomplexan species, e.g., in *T. gondii*, there are 22 singlet subpellicular microtubules [37] that are unusually stable and withstand high pressure, cold, and detergents, conditions incompatible with the survival for most microtubules [35].

In addition to microtubules, several apicomplexans possess another polymer form of tubulin. The conoid fibers resemble microtubules but their subunits are curled into an extremely tight coil, where tubulin is arranged into a polymer form that is different from typical microtubules. It is constructed of 10 to 14 curved tubulin sheets that form a hollow cone [34]. A central pair of intraconoidal microtubules of unknown function, which consists of 13 conventional protofilaments, runs down the middle of the conoid [34].

As I mentioned above, apicortin can be found mostly in species which are flagellated. Apicomplexans are in a special position in this respect. Because they are parasites, they do not need to move with flagella, so the flagella present in the common ancestor of the Alveolate were lost. However, at a certain stage of the life cycle, it can be found in male microgametes (although there are exceptions here as well, such as Piroplasmida and *Cryptosporidium*). Interestingly, this is the only stage in which the most characteristic organ of apicomplexans, the apical complex, is missing. Recent evidence suggests that “there was a remarkable contribution from the flagellum [38] (or the flagellar root apparatus [34]) to its evolution”. Due to the reduction of the flagellum to centrosomal centrioles in most apicomplexan cell stages, the apical complex is the most conspicuously retained element of the associated flagellar root structures [34]. Apicortin remained present even in Piroplasmida species, which possess neither flagellum nor conoid, except *B. microti*, whose minimal genome lacks many genes present in other apicomplexans.

## 9. The Role of Apicortin

There were some speculations concerning the role of apicortins in the earlier literature. Since both domains of apicortins are known to bind and stabilize microtubules, it was hypothesized that it is one of the MAPs (microtubule associated proteins) responsible for the enhanced stability of apicomplexan microtubular structures [1]. On the other hand, Hu et al. [39] partially purified the conoid of *T. gondii* and identified 59 proteins in this conoid-enriched fraction, including apicortin, denoted as 55.m08188. The authors suggested that these are good candidates for proteins restricted to the conoid/apical complex. Finally, it was suggested that the structural plasticity provided by the long, disordered N-terminus of apicortin may facilitate protein–protein interactions, which are necessary for the attachment to, and invasion of, host cells [5]. Recent development in the research of this protein seems to corroborate these ideas.

### 9.1. The Role of Apicortin in Toxoplasma gondii

#### 9.1.1. Structure

Apicortin is only localized at the conoid as shown by immunofluorescence staining of *T. gondii* and the labeling suggested that it is distributed all along the conoid fibers. The conoid in *T. gondii* is an about 380-nm-diameter motile organelle, consisting of 14 tightly curved and tilted tubulin fibers of nine protofilaments, which are non-microtubule polymers built from tubulin subunits and form a comma-shaped strip in cross-section [34,40]. The tubulin element of the conoid is made of the same α- and β-tubulin as the normal microtubules, so it is likely that tubulin-binding proteins specific to these fibers, e.g., apicortin, ensure this unusual conformation [34]. It would be obvious to study how tubulin polymerizes in the presence of apicortin, which has been attempted, but the fact that recombinant apicortin is insoluble in buffers used for tubulin polymerization excluded testing this process [28].

However, the group led by Murray and Hu demonstrated the role of apicortin (named as TgDCX by them) in shaping the structure of the conoid, in various indirect ways. For this purpose, they created apicortin-null mutant lines of *T. gondii* [40]. They found that tubulin disappeared or was present in a reduced amount in the apical complex of the null mutant, while canonical microtubules were intact [40]. Deletion of apicortin caused severe structural defects in the conoid. It was shorter, its shape was abnormal; however, some remnant conoid fibers were seen, which suggests that other protein(s) may also have a role in the stabilization of the conoid structure [40]. Other tubulin containing structures were unaffected. The defects were fully reversed by the expression of the apicortin coding sequence.

Another, very elegant method was used for the characterization of the interaction of apicortin and tubulin in a heterologous system [28]. A fluorescent protein-tagged apicortin was expressed in vivo in *Xenophus* eggs. It was found that apicortin not only binds tubulin polymers but also significantly modifies the organization of the microtubular network. Apicortin expression resulted in many short and smoothly curved microtubule fibers instead of normal microtubules, although they were much less sharply curved (mean radius of curvature is 4.7 μm) than the conoid fibers in *Toxoplasma* (0.25 μm). In contrast to normal microtubules, this apicortin coated fiber was resistant to nocodazole, a depolymerization agent.

As it was mentioned, it is an open question which other proteins participate in the shaping and stabilization of conoid. Conoid protein hub 1 (TgCPH1) [39,41], one of the candidates, was co-expressed with apicortin and both proteins were associated to the microtubules, further decreasing the radius of curvature (mean = 3.1 μm). However, when TgCPH1 was expressed alone, it was located in the cytosol. It suggests a “piggy-back” binding mechanism [42] for TgCPH1 via apicortin. Interestingly, in apicortin null-mutant parasites, this protein was targeted to the conoid, which suggests that further proteins take part in the interactions with the conoid [28]. On the other hand, apicortin was co-precipitated from the cytoskeletal fraction of *T. gondii* with dense granule protein 8 (GRA8), a component of the subpellicular cytoskeleton and that of the dense granules, which is secreted after host cell invasion [43]. These results show that apicortin influences the organization, shape, and stability of the tubulin polymers, and connects other conoid components to the tubulin core (Table 1).

This method was applied successfully to identify which regions of apicortin are important for the interaction with tubulin polymers [28]. By expressing various partial apicortin sequences in the *Xenophus* system, it was found that the DCX domain by itself is not sufficient for stable microtubule association but the partial p25α domain plus a part of the linker region binds to the microtubules. The removal of the N-terminal residues, preceding the partial p25α domain, had only a moderate effect. The significance of the N-terminus was corroborated in experiments when apicortin orthologs were expressed in the same system, the 3-3 *Chromera* and *Vitrella*, the *P. falciparum* and the *T. adhaerens* ones. Only the orthologs possessing a long N-terminal part preceding the partial p25alpha domain (*P. falciparum* apicortin and *Cvel_6797* from *C. velia*) were localized with microtubules. However, only unmodified *T. gondii* apicortin induced the curved microtubule arcs mentioned above.

#### 9.1.2. Function

The apicortin knockout parasites grew about four times slower compared with the wild-type ones. They were characterized concerning the parasite–host interaction as well. It was found that the host cell invasion was reduced by fourfold in comparison to the control. This effect can also be rescued by adding apicortin coding sequence. The exact mechanism of action is not known; it was speculated that “stabilization of the conoid fibers provides some beneficial mechanical advantage for host cell penetration or, alternatively, render the intraconoid passageway a better conduit for secretion from the parasite of the effector molecules known to be important during invasion” [40]. However, it is certain that the lytic cycle of the parasite is severely compromised due to the absence of apicortin resulting in defects in the conoid structure.

### 9.2. The Role of Apicortin in Plasmodium falciparum

*P. falciparum* belongs to the class Aconoidasida, which means “conoidless”, but a conoid-like structure was found in the ookinete stage of *P. gallinaceum* [46] and in another haemosporidian, *Leukocytozoon simondi* [47]. A conoid-specific protein, SAS6L, present in *T. gondii*, was also found in the sporozoite and ookinete stages of *Plasmodium berghei* [48]. The level of apicortin transcription relative to the whole transcriptome in *P. falciparum* is much higher in the gametocyte and ookinete stages compared with those in the erythrocytic life stages [28]. Thus, the possibility was raised that *P. falciparum* may possess a conoid-like structure at certain stages of its life cycle, and apicortin might have a role in its organization if such a structure exists [28,34].

#### 9.2.1. Localization

Immuno-localization of apicortin was monitored in various stages of *P. falciparum* [44,45]. Apicortin was localized on the surface of the parasite in the subpellicular region of both the trophozoites and the schizonts, where apicortin was co-localized with merozoite surface protein 1 (MSP1) and myosin A tail interacting protein (MTIP). In the merozoites, apicortin was observed at the apical end suggesting its role in apical complex formation. It was co-localized with both α- and β-tubulin. The localization of apicortin was confirmed in the nuclear and cytoplasmic extracts of the parasite with a major occurrence in the cytoplasm [44,45] (Table 1).

#### 9.2.2. In Silico and In Vitro Studies

Interaction of *P. falciparum* apicortin with *P. falciparum* tubulin was demonstrated by in silico docking studies, which have revealed that partial p25 domain residues present on the surface of apicortin are involved in the binding to both α- and β-tubulin [44]. It is worth noting that the partial p25alpha domain of *P. falciparum* is somewhat different from that of the other apicomplexans (cf. Figure 2). The role of this domain in the binding coincides with the finding by Leung et al. [28] who described the same for the apicortin-tubulin interaction in *T. gondii*. The binding was confirmed experimentally in vitro by immunological (immunoprecipitation, ELISA) and binding (surface plasmon resonance-SPR) methods as well as by a tubulin polymerization assay [44]. Interestingly, in contrast to *T. gondii* apicortin [28], the recombinant protein of *P. falciparum* expressed in *E. coli* could be successfully purified. Based on the SPR experiments, the strength of the binding was found in the submicromolar range: the K_d_ for the interaction of apicortin with α- and β-tubulin was 0.12 μM and 0.75 μM, respectively. Increased tubulin polymerization was found in the presence of apicortin, which can be disrupted by using tamoxifen, a drug that binds to apicortin. Tamoxifen effectively inhibited the growth of *P. falciparum* with an IC_50_ value of 8.3 μM; and defective progression of the parasite was observed. These effects were attributed to the disruption of apicortin–tubulin interaction.

#### 9.2.3. Effect of Human Erythrocyte microRNAs on Apicortin Expression and on Parasite–Host Interaction

Two human erythrocytic miRNAs, miR-150-3p and miR-197-5p, showed favorable in silico hybridization with both *P. falciparum* and *Plasmodium vivax* apicortins (Table 1). This finding was strengthened experimentally: co-expression of *P. falciparum* apicortin and either of these two miRNAs in a cell line model (HEK 293T cells) resulted in about 10-fold downregulation of apicortin at both the RNA and the protein levels [45]. A similar phenomenon was observed when mature schizonts were incubated with chemically synthesized mimics of miR-150-3p or miR-197-5p.

Based on these observations, these miRNA mimics were loaded into erythrocytes, and subsequently these erythrocytes were used for invasion by the parasite [45]. Apicortin expression was ~30% reduced in miRNA-loaded erythrocytes. Downregulation of apicortin by miR-197 resulted in loss of the organized structure of tubulin as demonstrated by the diffuse immunostaining of tubulin [44]. Growth of the parasite was hindered, and the rate of invasion decreased significantly. Some merozoites attached to erythrocytes but could not invade them. Micronemal secretion was also reduced, which was shown by the decreased level of the invasion-related microneme protein, apical membrane antigen 1. During these processes, the translocation of the two miRNAs from erythrocytes to *P. falciparum* was observed [45]. The results show that erythrocyte miRNAs might regulate the expression of certain parasite genes, as suggested earlier [49,50]. Thus, it was proposed that the mechanism of inhibition can be through translation repression of protein synthesis, due to the formation of a mRNA-miRNA hybrid [45].

## 10. Concluding Remarks

When apicortin was identified in silico, it was proposed that its function may be to stabilize specific cytoskeletal structures that are unique to apicomplexans [1]. It has also been hypothesized that it might have a role in parasite–host interactions [1,5]. Recent experimental results support both assumptions [28,40,44,45]. Apicortin has been shown to be localized in the apical complex; more precisely, in *T. gondii*, it is localized exclusively at the conoid, a tubulin-based structure that has an important role in host cell invasion [40]. Apicortin is essential for providing the correct structure and function of conoid. In *P. falciparum*, which has no conoid, apicortin is localized at the apical end; it has been suggested that it is involved in the formation of the apical complex [44]. In both species, downregulation of apicortin leads to impaired host cell invasion [28,45].

Apicomplexan species possess several hundred genes, which are specific for the phylum, and even more genes, which are absent in mammals/vertebrates but are present in these parasites. For example, apicomplexans have evolved hundreds of specialized invasion proteins, and contain lineage- and species-specific gene families, which are specialized for modulating host-specific adaptation [51]. Targeting parasite proteins that have crucial roles in these interactions is a key focus in the development of therapeutic agents against diseases caused by apicomplexan infection. Thus, apicortin, which seems to influence parasite–host interactions, is also a potential drug target. Since some mechanistic details of its function have been known, this protein has become more than a desirable target in drug development.

## Figures and Tables

**Figure 1 tropicalmed-06-00118-f001:**
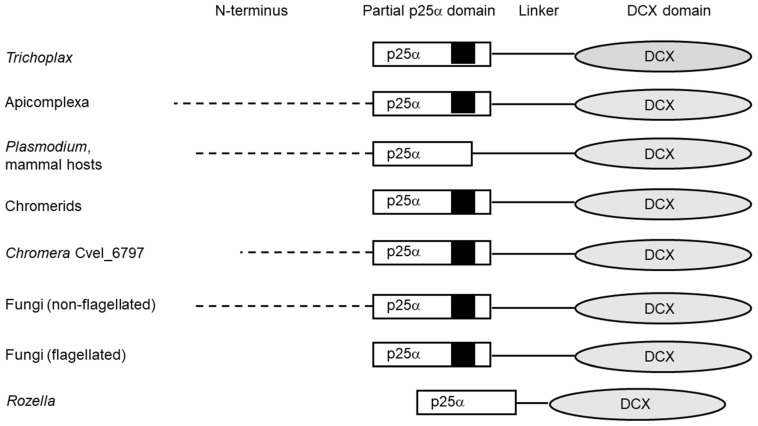
Schematic representation of apicortins. The partial p25alpha and doublecortin (DCX) domains are represented by rectangle and oval, respectively. The positions of the Rossmann-like motif (GXGXGXXGR) are indicated by black squares. The dashed line indicates the long unstructured N-terminus of apicortins.

**Figure 2 tropicalmed-06-00118-f002:**
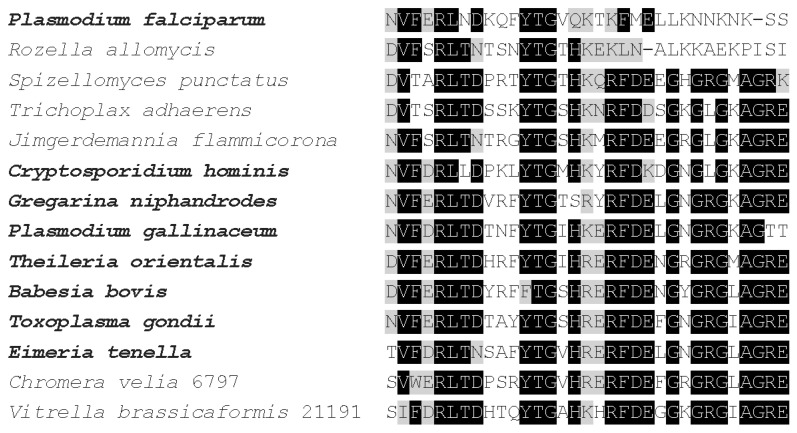
Multiple alignments of partial p25alpha domains of several apicortins by Clustal Omega [29]. Amino acids, which are identical and biochemically similar in the majority of the proteins, are indicated by black and gray shading, respectively. Bold letter indicates apicomplexan species.

**Figure 3 tropicalmed-06-00118-f003:**
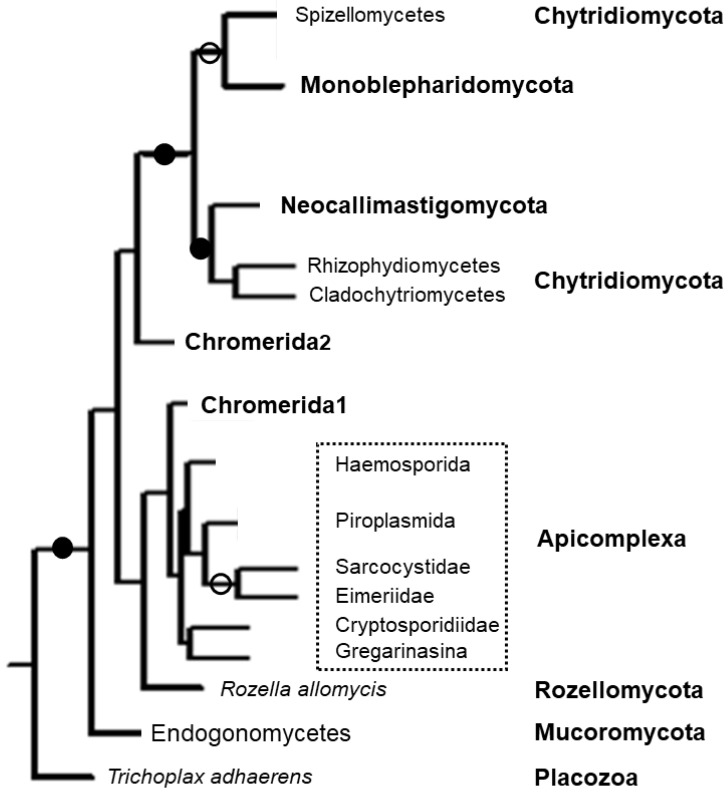
Phylogenetic tree of apicortins constructed by Maximum Parsimony (MP) analysis. Full and open circles at a node indicate that the branch was supported by MP bootstrap ≥85% and ≥50%, respectively. Bold, normal and italic letters indicate phyla, classes/orders/families and species, respectively. Phylogenetic units within the box belong to Apicomplexa. The figure based on Figure 3 of [8].

**Table 1 tropicalmed-06-00118-t001:** Interactions of apicortin at protein and RNA level.

Partner ^1^	Short Name	Type	Method	Reference
α- and β-tubulin		protein	immunofluorescence, immunoprecipitation, ELISA, surface plasmon resonance, tubulin polymerization, in silico docking	[44]
			fluorescence microscopy, co-expression in cell line model	[28]
Conoid protein hub 1	TgCPH1	protein	fluorescence microscopy, co-expression in cell line model	[28]
merozoite surface protein 1	MSP1	protein	immunofluorescence	[45]
myosin A tail interacting protein	MTIP	protein	immunofluorescence	[45]
dense granule protein 8	TgGRA8	protein	co-precipitation	[43]
tamoxifen	TMX	small molecule (drug)	surface plasmon resonance, cellular thermal shift assay, in silico docking, ELISA	[44]
miR-150-3p		microRNA	in silico hybridization, co-expression in cell line model	[45]
miR-197-5p		microRNA	in silico hybridization, co-expression in cell line model	[45]

^1^ The partners interact with apicortin protein except microRNAs, which interact with apicortin mRNA.

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
