# Peer review of "Apicortin, a Constituent of Apicomplexan Conoid/Apical Complex and Its Tentative Role in Pathogen—Host Interaction"

_tropicalmed, 2021, doi:10.3390/tropicalmed6030118_

Round 1
Reviewer 1 Report
This review is an excellent summary of the current status on the research into apicortin that is emerging as an important protein in apicomplexan parasite-host interaction and hence an important drug target. The review is by a key researcher in this field and therefore an essential contribution to this special issue.
In the attached annotated copy, a few editorial changes have been suggested to make the meaning clearer and in some instances clarification is sought from the author. Phylogenetic tree and its Figure legend requires additional information to be more informative as indicated.
It is suggested that a schematic of apicortin structure be provided and a diagram that summarises the key interactions with e.g. tubulin as described in the text.

Author Response
Thank you for your sympathetic criticism and valuable suggestions. I accepted most of them, including the style and grammar changes. The changes are labeled by yellow background in the revised version.
The phylogenetic tree and its legend were complemented.
A schematic of the apicortin structure was shown in Figure 1. I cannot present a more detailed picture since only the structure of the DCX domain is known.
A table was introduced that summarizes the interactions of apicortin.
Reviewer 2 Report
The manuscript reviews the Apicortin protein and its role in Apicomplexa parasites, particularly Plasmodium falciparum and Toxoplasma gondii.
The work is well-conducted, structured, and clearly written.
The topic is interesting and worthy of investigation for the possible role of this protein as a target for the treatment of these medically important diseases.
Therefore, I do recommend the acceptance of the article in its present form.
Author Response
Thank you for reading my manuscript and for your sympathetic criticism.
Reviewer 3 Report
The author of the manuscript has written a lovely review, below; I indicate my doubts and suggestions:
- The author in lane 112 mentions: The partial p25alpha domain is the most conservative part of the protein, and there are two exceptions: R. allomycis, and Plasmodium orthologs in species infecting mammals (but not birds), which looks good if we see the Figure 1.
However, in Figure 2: The author shows an alignment (of partial p25alpha domains of several apicortins) which include Plasmodium falciparum (human malaria) and Plasmodium gallinaceum (bird malaria). And create confusion. Could the author please clarify it?
- Also, maybe the author could mention which Plasmodium species were analyzed because more than 200 species of Plasmodium’s can infect different hosts, including humans (5 malaria species), birds, bats, lizards, and antelopes.
- Plasmodium species was not included in Figure 3 (Phylogenetic tree of apicortins). is it because all Plasmodium species are considered inside haemosporida order/suborder? (Table S1).
- The author did not mention the hypnozoite stage in the subtitle: 7—Life stages of apicomplexans. The author cited only one website. The paragraph needs more references cited.
- Babesia microti is cited in the manuscript, but Babesia bovis (p25alpha domain sequence) was used for the alignment (Figure 2). Both Babesia species had identical sequences (is conserved?). These small changes create some doubts in readers.
- Cosmetics changes:
Double comma in lane 250.
Quotation mark in lane 252.
Figure 3. All names are “underlined” in red color.
Author Response
Thank you for your sympathetic criticism and valuable suggestions. The changes are labeled by yellow background in the revised version.
My detailed answers are as follows:
- I wrote in lines 112-115: “The partial p25alpha domain, in general, is the most conservative part of the protein (Fig. 2), however, there are two exceptions. R. allomycis and Plasmodium orthologs present in species infecting mammals (but not birds) lack the final part of this domain, which in-cludes the Rossmann-like motif (Fig. 2).” Fig. 2 shows, in the first two lines, that R. allomycis and Plasmodium falciparum are exceptions, indeed, since the last 11 amino acids of their partial p25alpha domain differ completely from the corresponding amino acids of the other orthologs.
- When I mention Plasmodium it corresponds to the whole genus except otherwise stated. I carefully checked the text, and I found that in three cases I did not mention the species. I completed the names: line 342, P. gallinaceum, line 344 P. berghei, line 395 P. falciparum.
- Yes. Apicomplexan species are not listed in Fig. 3, but higher phylogenetic units (orders, families) are shown. Indeed, Plasmodium species are within the order Haemosporida.
- A paragraph about the life stages of apicomplexans was included into the paper because some of the papers reviewed discussed intensively the occurrence of apicortin in the various life stages of Plasmodium falciparum thus it was necessary to give a short overview. Thus, beside the general features of the life stages of the apicomplexans (sporogony, merogony and gametogony), the stages of the Plasmodium life cycle were detailed. The hypnozoite stage was not mentioned since it is absent in P. falciparum. However, I accepted the suggestion and added the hypnozoite stage (lines 205-207 and a new reference, Ref. 31). An additional reference about the life stages of Plasmodium was also added (Ref 32, Baton et al. 2005).
- Babesia microti does not contain apicortin (lines 31, 184) thus it was not used in the multiple alignment.
- Corrected.